# Respiratory Adsorption of Organic Pollutants in Wastewater by Superhydrophobic Phenolic Xerogels

**DOI:** 10.3390/polym14081596

**Published:** 2022-04-14

**Authors:** Yinchun Li, Depeng Gong, Youliang Zhou, Chaocan Zhang, Chunyang Zhang, Yitian Sheng, Shu Peng

**Affiliations:** School of Materials Science and Engineering, Wuhan University of Technology, Wuhan 430070, China; liyinchun@whut.edu.cn (Y.L.); gdpyy1@163.com (D.G.); zyl_2013@whut.edu.cn (Y.Z.); 13476089628@163.com (C.Z.); chrysemy@163.com (Y.S.); ps15771189488@163.com (S.P.)

**Keywords:** high selectivity, compressibility, surface modification, organic contaminants, adsorption, recyclability

## Abstract

Organogel adsorbents are widely used for the adsorption of hard-to-degrade organic pollutants in wastewater due to their natural affinity to the organic phase in water. In this study, phenolic xerogels (PF) synthesised in the ethylene glycol inorganic acid system are used as a backbone and superhydrophobic phenolic xerogels (ASO-PF) are obtained by grafting aminosilanes onto the PF backbone via the Mannich reaction. The modified ASO-PF not only retains the pore structure of the original PF (up to 90% porosity), but also has excellent superhydrophobic properties (water contact angle up to 153°). Owing to the unique pore structure, ASO-PF has excellent compression properties, cycling 50% compression deformation more than 10 times without being damaged, with a maximum compression deformation of up to 80%. A squeeze–suction–squeeze approach is proposed for selective adsorption of organic pollutants in homogeneous solutions based on the recyclable compression properties of ASO-PF. The ASO-PF is put under negative pressure by squeezing, and when the pressure is released, the adsorbed liquid enters the ASO-PF, where the organic pollutants are retained by the adsorption sites in the skeleton, and then the remaining water is discharged by squeezing. This breathing ASO-PF holds great promise for organic pollutant adsorption and recovery applications.

## 1. Introduction

As industrialization progresses, many poisonous and damaging compounds in industrial production, such as organic solvents [1], dyes [2,3], and phenolic pollutants [4], are inexorably expelled into nature with wastewater, resulting in environmental contamination [5]. Common water treatment methods used in the field of wastewater treatment include chemical oxidation [6], ion exchange [7], biodegradation [8], and adsorption [9]. A class of organic dyes and phenolic pollutants exist in wastewater, and these aromatic pollutants are difficult to remove directly from water or are chemically oxidized and biodegraded, even at very low concentrations still possessing a biocidal effect and thus endangering human health [10,11,12,13]. For these hard-to-degrade pollutants, traditional water treatment methods are inadequate to cope with modern wastewater treatment technologies [14]. Chemical adsorption methods can effectively adsorb these organic pollutants due to their high efficiency, low cost, and simple operation [15]. In recent years, gel-based adsorbents have been widely employed in the treatment of organic pollutants [16]. Aerogels and xerogels are widely applied as adsorbents for organic dyes and phenolic pollutants due to their low density and high porosity [17,18]. Although aerogels have a higher adsorption capacity, they are complex and expensive to synthesise [19]. At present, the most mainstream industrial production route of aerogel is to adopt organosilicon source (such as orthosilicate) as raw material combined with supercritical drying process for production. The high price of organosilicon source as raw material and supercritical equipment are special equipment with manufacturing threshold and high investment, which makes the cost under pressure. Xerogels are obtained by direct removal of the solvent from wet gels without exchange with supercritical fluids, offering greater economic benefits.

Organic xerogels are porous frameworks formed by the drying of polymer networks swollen in organic solvents [20]. With their adjustable pore structure and natural affinity to the organic phase in water, organic xerogels have been widely used for the adsorption of organic pollutants from wastewater. Ren et al. [21]. prepared an ultra-fast self-healing organ gel based on dynamic covalent chemistry using poly(vinyl alcohol)(PVA), 4-formylphenylboronic acid (FPBA), and dihydrazide tartrate (TDH). Their freeze-dried xerogels selectively adsorbed methylene blue from water by π-π stacking and electrostatic interactions with a removal efficiency of 95.89%. Hernández-Abreu et al. [22] compared the adsorption capacity of synthetic resorcinol–formaldehyde carbonised xerogels (RFX), lignin-based activated carbon (KLP), and commercial activated carbon (F400) for phenol. The KLP and RFX adsorbents exhibited faster kinetic adsorption because of their more open porous structure.

The chemical and physical structure of the adsorbent surface is well recognized to have an impact on the adsorption process [23,24]. The surface functionalization of xerogels have been shown in numerous studies to be an effective approach for adsorbing different contaminants in water. Sriram [25] modified the surface of diatomite with mesoporous silica xerogel and introduced amine groups to investigate its dye adsorption ability in water. The dye adsorption effectiveness of the planned xerogel increased from 58% to approximately 99% after the addition of amine groups. Ptaszkowska-Koniarz [26] added an amine group to the surface of a resorcinol–formaldehyde carbon xerogel and then impregnated it with copper chloride, resulting in a carbon xerogel that could adsorb caffeine in water. The adsorption effect of caffeine is optimum when the amine group and copper chloride coexist, and the maximal adsorption capacity can reach 91–118 mg·g^−1^.

Phenolic porous materials are an important porous material that is developing rapidly in materials science today. The delicate chemical structure gives phenolic porous materials excellent properties such as low flammability, high dimensional stability, and excellent solvent resistance [27]. Resorcinol and formaldehyde are often used as raw materials in phenolic gels, but the high cost of resorcinol and the high storage requirements limit their utilization in practical production. We synthesize the phenolic gels by generating the methyl-order phenolic resin directly, then dissolving it in ethylene glycol using the sol–gel method with acid curing to produce a wet gel. This strategy is more feasible in terms of production. Due to changes in solvent-gel interfacial tension, direct drying of wet gels during solvent removal causes shrinkage of the entire skeletal structure, leading to greater density and decreased porosity [28]. By lowering the capillary pressure between the skeletons, a more advanced drying approach such as freeze-drying can suppress interfacial tension and thereby reduce shrinkage. We discovered that low-density phenolic xerogels made with a glycol inorganic acid system have outstanding flexibility and compressibility, suggesting that they could be used in wastewater treatment. Aminosilanes with low ammonia values were selected as raw materials and grafted onto the PF xerogels by the Mannich reaction [29], resulting in excellent selectivity throughout the backbone. As the whole backbone of the prepared ASO-PF xerogels is superhydrophobic, homogeneous aqueous solutions containing low concentrations of organic contaminants cannot enter the interior of the xerogels. In order to better utilize the adsorption sites in the internal pores of the modified xerogels, a respiratory adsorption method was proposed to adsorb organic pollutants in homogeneous wastewater based on the compressibility of the prepared ASO-PF xerogels. Firstly, the ASO-PF xerogel is squeezed in the adsorbed liquid to make it in a negative pressure state, and after releasing the pressure, the adsorbed liquid enters the interior of the xerogel, where the organic pollutants are captured by the adsorption sites in the internal pores. Subsequently, the non-adsorbed water is mechanically squeezed out, and the cycle of squeezing–suction–extrusion can effectively separate various organic solvents, dyes, and phenolic pollutants in the adsorbed water. The respiratory adsorption method of this superhydrophobic porous material has broad application space in the field of wastewater treatment.

## 2. Experimental Section

### 2.1. Materials

The laboratory manufactured the methyl phenolic resin, Dow Corning supplied the amino silane 8040A, and Sinopharm Chemical Reagent Co. provided the other raw ingredients.

### 2.2. Experimental Preparation

#### 2.2.1. Synthesis of Methyl-Order Phenolic Resin

Phenol and paraformaldehyde preheated in advance were added to the three-mouth flask according to the aldehyde-to-phenol ratio of 1.8 and pre-reacted for 3 h at 60 °C. A certain amount of alkali catalyst NaOH was added every 1 h (the total amount of catalyst was 1% of the substance of phenol), and then the temperature was slowly increased to 85° to continue the reaction for 1.5 h to obtain the methyl-order phenolic resin with a water-solubility ratio of 10:12.

#### 2.2.2. Preparation of Phenolic Xerogel (PF)

Using the sol–gel method, 4 g of alpha phenolic resin was dissolved in 40 g of ethylene glycol, stirred magnetically for 30 min, 2 g of dilute hydrochloric acid (37%) was added as the catalyst and stirred for 30 min, and then put into the oven at 70 °C for 24 h to obtain the wet gel. The wet gel was solvent replaced with distilled water for three days, and the water was changed every 8 h, and the phenolic xerogel skeleton was obtained by freeze-drying.

#### 2.2.3. Amino Silane Modified Xerogel (ASO-PF)

Ultrasonically disperse a certain amount of amino silane in tetrahydrofuran for 30 min, then add equimolar formaldehyde solution (37%), mix well, add a certain amount of PF xerogel, adjust the solution pH to weak acidity, and finally bake at 40 °C for 6 h. The reacted ASO-PF was washed with hexane solution to remove the residual amino silane before being dried at room temperature to produce a hydrophobic xerogel.

### 2.3. Characterization

To analyse the molecular structure of the samples, Fourier transform infrared spectroscopy (FTIR) testing was performed using a Thermo Nicolet 6700 spectrometer. X-ray photoelectron spectroscopy (XPS) with an Axis Supra energy spectrometer to analyse the sample’s surface elemental composition and charge correction of the data. The surface and cross-sectional morphology of the samples were observed using a cold field emission scanning electron microscope with an additional X-Max N80 energy spectrometer (SEM) model JSM-7500F. The cyclic compression properties of the samples were determined using an Instron 5967 electronic universal material testing machine. A contact angle analyser is used to measure the sample water contact angle (WCA). Using an Essence 754PC UV-Vis spectrophotometer, check the sample’s UV-Vis absorption spectrum. An AutoPore lv 9510 high performance fully automated mercury piezometer was used to measure the pore size distribution of the samples. Nitrogen adsorption–desorption isotherms were measured using an ASAP 2020 fully automated specific surface area and porosity analyser. The absolute density of the sample was determined using the AccuPyc 1330 fully automated true density analyser and the porosity was calculated using the following Equation (1).
(1)$$Porosity\%=\left(1-\frac{Apparent\;density}{True\;Density}\right)\times 100\%$$

### 2.4. Oil–Water Separation Test

In the static adsorption test, 0.1 g of ASO-PF was immersed in various organic solvents and oils (including hexane, xylene, toluene, chloroform, methylene chloride, ethyl acetate, petroleum ether, liquid paraffin, etc.), and the adsorption test was repeated after saturation adsorption by removing the surface oil and organic solvents with filter paper for weighing. Static adsorption Q = (m_1_ − m_0_)/m_0_, where m_1_ and m_0_ represent the mass of the xerogel before and after adsorption. Dynamic adsorption experiments used xerogels adsorbed with Sudan red-stained hexane on water and Sudan red-stained methylene chloride underwater to simulate oil sorption on water and oil sorption underwater.

### 2.5. Adsorption of Phenol

To investigate the adsorption mechanism and capacity of the modified xerogel on phenol, ASO-PF was used as the adsorbent and phenol as the adsorbent mass in phenol adsorption experiments. To begin, the absorbance of phenol solutions at various concentrations was measured using a UV spectrophotometer at a maximum wavelength of 270 nm [30], and the phenol standard curve was plotted using Lambert’s law [31], as shown in Appendix A, with the correlation coefficient R^2^ = 0.998. The same mass of ASO-PF was used to adsorb 100 mg·L^−1^ of phenol solution with different pH values in a conical flask to study the effect of pH value on the adsorption performance of ASO-PF on phenol. With a contact time of 1 min-24 h and shaking at room temperature of 25 °C, 0.1 g of xerogel was added to 20 mL of a predetermined concentration of phenol solution. 

Isothermal adsorption curves were plotted to investigate the effect of time on adsorption performance. Calculate the amount of phenol adsorbed by ASO-PF using Equations (2) and (3).
(2)$$Adsorption\;rate\%=\frac{C_{0}-C_{e}}{C_{0}}\times 100\%$$
(3)$$Q_{e}=\frac{\left(C_{0}-C_{e}\right)V}{M}$$
where *C*_0_ and *C_e_* (mg·L^−1^) represent the initial phenol solution concentration and phenol solution concentration after adsorption equilibrium, *V* (mL) represents the volume of adsorbed liquid, *M* (g) represents the mass of adsorbent, and *Q_e_* (mg·g^−1^) represents the adsorption amount of phenol by ASO-PF.

The isothermal model of Langmuir and Freundlich was used to build isothermal adsorption curves to examine the mechanism of ASO-PF interaction with phenol solutions based on adsorption experimental data [32,33].

The Langmuir model assumes that the adsorbent surface is homogeneous surface consists of the same adsorption sites and monolayer adsorption occurs on the adsorbent homogeneous surface [34]. The Langmuir isothermal adsorption model is expressed as follows (4):(4)$$Q_{e}=\frac{Q_{max}K_{L}C_{e}}{1+K_{L}C_{e}}$$
where *Q_e_* (mg·g^−^^1^) denotes the amount of phenol adsorbed per unit mass of adsorbent, *C_e_* (mg·L^−1^) denotes the solubility of the phenol solution at equilibrium, and *K_L_* (L·mg^−1^) and *Q_max_* (mg·g^−1^) denote the Langmuir parameters relating to adsorption rate and capacity, respectively.

The Freundlich model assumes that the adsorbent surface is non-homogeneous. It assumes that the adsorbate first occupies the stronger adsorption sites while the binding strength decreases as the adsorption sites are occupied [35]. Its expression is as follows (5):(5)$$Q_{e}=K_{F}C_{e}{}^{1/n}\;$$
where *K_F_* is the Freundlich parameter related to the adsorption capacity and *n* denotes the adsorption strength.

Adsorption kinetics defines how quickly or slowly an adsorption process occurs. The two most widely utilized adsorption kinetic models are the quasi-first-order kinetic model and the quasi-second-order kinetic model [36]. 

The quasi-first-order kinetic model assumes that the adsorption rate is mostly driven by the diffusion resistance of the adsorbent molecules within the adsorbent, and that there is a linear relationship between the adsorbent concentration and the adsorption rate [37]. The following are the expressions:(6)$$ln\left(Q_{e}-Q_{t}\right)=lnQ_{e}-k_{1}t$$
where *Q_e_* (mg·g^−1^) is the adsorption amount of adsorbent at adsorption equilibrium, *Q_t_* (mg·g^−1^) is the adsorption amount of adsorbent at time *t*, and *k*_1_ (min^−1^) is the adsorption rate constant.

The quasi-second-order kinetic model assumes that chemisorption has a strong influence on the adsorption rate and is related to the electronic interactions between the adsorbent and the adsorbate [38], which is expressed as follows.
(7)$$\frac{1}{Q_{e}-Q_{t}}=\frac{1}{Q_{e}}+k_{2}$$
where *k*_2_ (min^−1^) is the adsorption rate constant.

### 2.6. Dye Adsorption

The adsorption of Congo red stain in water was used as an example to fix the concentration of the adsorbed dyes and study the visualization of the PF and ASO-PF adsorption behaviour on dyes in wastewater.

## 3. Results and Discussion

### 3.1. Material Synthesis and Structural Characterization

In this paper, PF xerogels were synthesized using resole phenolic resin as raw material. The molecular chain of alpha phenolic resin contains many active phenolic hydroxyl groups, and the neighbouring para-hydrogen on the benzene ring can also undergo nucleophilic substitution, providing a favourable reaction site for hydrophobic modification. The phenolic wet gel was synthesized by using the sol–gel method on the methyl-order phenolic resin in ethylene glycol and curing with catalyst, while the PF xerogel was prepared by replacing the solvent and freeze-drying. By using the Mannich reaction, the modifier amino silane is grafted onto the surface of PF xerogel, as shown in Figure 1. In the acidic environment, formaldehyde undergoes carbonyl protonation, and the amino group on the amino silane undergoes nucleophilic addition to the carbonyl group to form an imine ion intermediate to attack the para-active hydrogen on the benzene ring, and the amino silane is grafted onto the PF xerogel backbone to achieve the purpose of hydrophobic modification.

The SEM pictures of the xerogels are shown in Figure 1, and all samples display a three-dimensional porous structure. The PF xerogel in Figure 1a–c is consisted of a stack of micron-sized spheres, with minor fusion between them providing some skeleton strength. When combined with the pore size distribution diagram (Figure 2b), the pore size distribution of PF xerogel is around 10um, and the bigger pore structure is useful in reducing the contraction and collapse of the entire xerogel due to capillary pressure during the freeze-drying process. The synthesized PF xerogels also have low density (~0.113 g·cm^−3^) and large porosity (~92.0%) (Table 1), which confer good adsorption properties. The amino silane was grafted onto the phenolic xerogel backbone, and the modified xerogel still maintained a good pore structure. As shown in Figure 1d–f, the modifier amino silane was wrapped around the entire surface of the PF xerogel backbone, which indicates that the hydrophobic modification was not only present on the surface of the xerogel, but also its internal pore size was modified, making both the surface and internal pores of the PF xerogel highly selective. The improved ASO-PF density increased marginally from 0.113 g·cm^−3^ to 0.137 g·cm^−3^, while the porosity remained over 90%. (Table 1). The N_2_ adsorption-desorption isotherm of ASO-PF is depicted in Figure 2a. According to the IUPAC classification [39], the isothermal adsorption curve of ASO-PF is type II, which represents the typical physical adsorption process of macroporous adsorbents. Due to the strong interaction between the adsorbent and the surface, the adsorption rises rapidly at lower relative pressures and the curve is convex. The isotherm inflection point usually appears near the monolayer adsorption, and the multilayer adsorption gradually forms as the relative pressure continues to increase.

The successful hydrophobic alteration of the entire ASO-PF framework was validated by FTIR analysis of modified and unmodified xerogels. The FTIR spectrum revealed several new absorption peaks after being modified with amino silane (Figure 3a). Among them, the anti-symmetric stretching vibration peak of Si-O-Si bond and the planar rocking vibration characteristic absorption of Si-(CH_3_)_2_ appear at 1097 cm^−1^ [40], and the symmetric stretching vibration peak of Si-O-Si bond is around 801 cm^−1^ [41]. The deformation vibration characteristic absorption of -CH3 of Si-(CH_3_)_2_ is at 1261 cm^−1^ [42]. The FTIR results confirmed the strong interaction between the amino silane and the PF xerogel, indicating that the xerogel and the amino silane form a new covalent bond, which is beneficial in practical applications to maintain a long and stable hydrophobicity. XPS analysis confirmed the interfacial bonding of the amino silane to the PF xerogel. As shown in Figure 3b, three new bands appear in the amino silane-modified xerogel, corresponding to the characteristic peaks of N1s (400 eV), Si2p (103 eV), and Si2s (150 eV), respectively [43]. The high-resolution XPS spectra confirmed the signals of C-N and C-Si chemical bonds. The C1s spectrum of the modified xerogel is divided into three main component peaks, with 284.78 eV representing C on the benzene ring, 286.08 eV indicating the C-OH and C-N bonds, and 284.18 eV indicating the peak position of the C-Si bond [44]. The peak position of 400.3 eV in the N1s spectrum further supports the presence of the C-N bond at the interface between the amino silane and the xerogel bulk, indicating that the amino silane has been successfully grafted onto the surface of the PF xerogel.

### 3.2. Compression Performance of PF and ASO-PF

During the exploration of the whole subject, as shown in Appendix A, the compatibility of the glycol inorganic acid system with the resin was good and the PF xerogels synthesised from both the mixed and single glycol systems were elastic and became more elastic as the percentage of glycol increased. The modified ASO-PF still has good elasticity and compressibility properties. This could be due to the compressibility of the entire xerogel presence of ether bonds and a large deformation space, as well as the large internal particles of the PF xerogels synthesized by the glycol inorganic acid system and fusion, which gives the backbone some strength. The stress–strain curves of the maximum compressive deformation of PF and ASO-PF are shown in Figure 4a. The original PF xerogel synthesized by the glycol system can withstand 85% of the maximum compressive deformation, while the modified ASO-PF can still withstand 80% of the maximum compressive deformation, as shown in the figure. The stress–strain curves for various compressive deformations also confirm that PF and ASO-PF deform similarly when subjected to compressive stress (Figure 4b,c). The ASO-PF was put through a loop compression test with a 50% deformation (Figure 4d). The ASO-PF skeleton was loose during the first compression and had a large deformation space attributed to the presence of a large pore structure. The stress–strain of ASO-PF stabilized after the second compression, and it was able to withstand 10 compression cycles without being damaged.

### 3.3. Oil–Water Separation Performance of the PF and ASO-PF

The material’s wetting behaviour is critical for wastewater treatment. The water contact angle (WCA) measurements of the xerogels before and after modification are shown in Figure 5a,b. Within 1 s, water droplets on the surface of the PF before modification spread and penetrated directly into the sample’s interior. Without hydrophobic properties, the water contact angle was 0°. The modified ASO-PF surface displayed a nearly circular water droplet with a contact angle of 153° and remained in this state without penetrating the xerogel interior, demonstrating stable superhydrophobic properties. The hydrophobic properties of the ASO-internal PF’s profile were measured after the ASO-PF was cut. The trimmed xerogel profile retains good hydrophobic properties and a contact angle of around 149°, as shown in Figure 5c. It is demonstrated that the modification occurs not only on the surface, but also the pores inside the xerogel, which greatly increases the selectivity of the modified ASO-PF.

Organic solvents and petroleum pollution are the major contributors to industrial wastewater and marine pollution in environmental management. Different organic solvents were selected as target organic pollutants to evaluate the adsorption capacity of ASO-PF xerogels. For this two-phase wastewater solution, direct static adsorption can be collected. The adsorption capacity of the modified ASO-PF was up to 13 times its own mass under static adsorption, as shown in Figure 6g, and remained stable for several cycles (Figure 6h). Even at lower densities, the adsorption capacity of n-hexane could reach 5.3 g·g^−1^, and the dynamic adsorption experiments of ASO-PF were characterized by adding drops of Sudan red-stained n-hexane and dichloromethane to water, respectively. ASO-PF rapidly adsorbed hexane on the water surface and dichloromethane beneath the water, as shown in Figure 6a–f. The xerogel exhibits the silver mirror phenomenon in water, and due to its superhydrophobic property, there is a water film wrapped on the surface of the xerogel when it is clamped deep into the water with tweezers, achieving the purpose of isolating the surrounding water and allowing it to maintain a long and stable hydrophobic state in water. To verify this conclusion, an adsorption–desorption experiment was conducted on ASO-PF, and the xerogel was saturated with n-hexane and then dried for 10 cycles to obtain the cyclic contact angle Figure 6i. After 10 cycles of adsorption-desorption, the xerogel could still maintain a water contact angle of about 150°, achieving a stable superhydrophobic effect, which is conducive to reuse in the field of wastewater treatment.

### 3.4. Adsorption Capacity of ASO-PF on Phenol

In addition to insoluble organic solvents, industrial wastewater contains a large number of phenolic compounds that are soluble in water. These phenolic compounds (e.g., phenol) are more difficult to remove than normal organic substances. Additionally, for superhydrophobic porous materials, it is difficult for low concentration phenol solutions to enter the interior of the porous materials due to the water-repelling effect of the surface, making phenol only captured by the adsorption sites on the surface, which leads to the decrease in the adsorption efficiency. Therefore, a respiratory adsorption approach is proposed to improve the adsorption efficiency of pollutants in homogeneous phenol solutions by combining the recyclable compression properties of ASO-PF xerogels. The low concentration phenol solution is made to enter the interior of ASO-PF by squeeze–inhalation–squeeze, and then the phenol is captured by the adsorption sites in the internal pores, and the remaining water is squeezed out to achieve the separation effect. The adsorption capacity and adsorption behaviour of ASO-PF on phenol in wastewater are shown below.

#### 3.4.1. Effect of pH Value on the Phenol Adsorption Capacity of ASO-PF

Xerogel adsorption of phenol is synergistic in numerous ways. pH, for example, is a crucial influencing factor in the electrostatic interaction between the adsorbent and the adsorbed substance. It not only impacts the adsorbent surface’s potential, but also determines the type of adsorbate in the solution. Figure 7a shows that the adsorption capacity of the modified xerogel on phenol rises with increasing pH before stabilizing. This is due to the fact that under acidic conditions, on the one hand, the hydrogen of phenol hydroxyl groups on phenol is not easily ionized and phenol exists mainly as C_6_H_5_OH in aqueous solution, and the adsorption of phenol by ASO-PF is mainly physical adsorption. On the other hand, the presence of a large amount of H^+^ in the solution competes with phenol for the adsorption sites of ASO-PF leading to the decrease in the adsorption performance of the adsorbent on phenol. The adsorption rate reaches the maximum when pH = 7. The adsorption rate of phenol decreases but stabilizes when the pH value continues to increase. This is due to the fact that at neutral or weak alkalinity, phenol exists in aqueous solution mainly as C_6_H_5_O^-^, which is electrostatically attracted to the cations on the surface of ASO-PF with the introduction of nitrogen-containing groups. The synergistic effect of physical and chemical adsorption improved the phenol adsorption by ASO-PF at this point.

#### 3.4.2. Adsorption Kinetics and Adsorption Isotherms of ASO-PF

The adsorption kinetic curves of ASO-PF in phenol solution with an initial concentration of 100 mg·L^−1^ are shown in Figure 7b. In the first two hours, ASO-PF has a very high adsorption efficiency. During this period, phenol in the aqueous solution rapidly occupies the chemisorption sites on the surface of ASO-PF, after which the binding sites tend to be saturated. Phenol continues to diffuse along the internal pore surface, using the surface pores as channels, increasing the length and resistance of mass transfer and the adsorption efficiency slowly. The adsorption efficiency could reach 88% after 24 h. According to the fitting results of the kinetic adsorption studies, the adsorption of phenol by ASO-PF is more inclined to the quasi-secondary kinetic model, implying that the adsorption is mostly based on chemisorption. Because of the existence of a significant number of macropores in the produced ASO-PF xerogel, which contributes little to the specific surface area, phenol adsorption by ASO-PF at rest is mostly based on chemical site adsorption. Since ASO-PF has good compression performance, the phenol-containing wastewater can be continuously extruded and discharged by the cyclic extrusion–inhalation–extrusion method of respiratory adsorption, which can more effectively utilize the adsorption sites in the pores inside the ASO-PF xerogel. The isothermal fitting curve is illustrated in Figure 7c, where the initial phenol concentration has a substantial effect on ASO-PF adsorption. Adsorption of ASO-PF increased with concentration, from 12.21 mg·g^−1^ (100 mg·L^−1^) to 77.35 mg·g^−1^ (500 mg·L^−1^). The correlation coefficient of the Langmuir model was 0.999 based on the fitted values. The phenol adsorption by ASO-PF was more consistent with the Langmuir model, indicating that the adsorption process was unimolecular layer adsorption. Although the adsorption amount of modified ASO-PF rose from 46.84 mg·g^−^^1^ (500 mg·L^−1^) to 77.35 mg·g^−1^ compared to the unmodified PF xerogel, the adsorption of phenol by ASO-PF was low and far from saturation, as shown in Figure 7c. Perhaps it is because the superhydrophobic characteristics of ASO-whole PF’s skeleton prevent continuous diffusion of phenol into the interior pores at lower concentrations of aqueous phenol solutions. The aqueous phenol solution cannot penetrate the interior of the xerogel and is trapped by the surface pores, resulting in a decrease in adsorption. The surface energy of the adsorbent was further lowered by increasing the phenol concentration in order to allow the adsorbent to permeate into the ASO-PF xerogel and better use the selectivity inside the ASO-PF. 

Figure 7e shows that when the phenol aqueous solution concentration reaches 12.5 g·L^−1^, the surface energy is comparable to that of ASO-PF. Droplets of the adsorbed solution seem hydrophobic when they initially drop onto the xerogel’s surface and then slowly penetrate into the xerogel’s interior. The adsorption curves of the highly concentrated phenol solution in Figure 7d further confirm the significant effect of increasing concentration on enhancing the adsorption of ASO-PF. At 12.5 g·L^−1^, the phenol solution started to enter the interior of the ASO-PF xerogel, which continued to preferentially adsorb phenol in water due to the high selectivity of the entire ASO-PF skeleton. The utilization of the whole xerogel skeleton led to a significant increase in the adsorption capacity, jumping from 0.7427 g·g^−1^ (point A) to 1.188 g·g^−1^ (point B), after which the growth trend gradually slowed down. It shows that ASO-PF has better adsorption effect and adsorption capacity for high concentration phenol solution.

#### 3.4.3. Adsorption of Organic Dyes by ASO-PF

In order to better show the respiratory adsorption mode of the xerogels to organic pollutants in homogeneous solutions before and after modification, the selectivity of PF and ASO-PF xerogels to dyes and water was investigated using Congo red stain as an example. As shown in Figure 8a–c, when the unmodified PF xerogel adsorbed the dye, Congo red and water were adsorbed into the interior of the xerogel without any difference. Since the PF xerogel itself is composed of organic skeleton, it has a certain adsorption capacity to the organic phase in water. When the PF xerogel is compressed, the water inside is extruded, and the Congo red is trapped in the organic skeleton, and the extruded water is colourless and transparent. From the wetted filter paper underneath the xerogel, it can be seen that there is basically no residue of Congo red stain in the extruded water. The modified ASO-PF xerogel was placed in the same quantity of Congo red dye, and owing to its superhydrophobicity, ASO-PF absorbed Congo red dye in water preferentially. Congo red dye was continually absorbed into the inside of the ASO-PF xerogel and trapped by the internal pores by continuous extrusion and release. Water is evacuated with the extrusion due to the hydrophobic characteristics of the whole skeleton. Figure 8d–g also show that as the ASO-PF is constantly compressed and released, the Congo red dye is gradually absorbed and the solution becomes colourless and transparent. Simultaneously, ASO-PF continues to float on the liquid surface after numerous cycles of extrusion beneath the liquid surface, demonstrating ultra-hydrophobic properties. To recycle the dye on the surface of the adsorbed ASO-PF xerogel, immerse it in acetone.

## 4. Conclusions

In this article, compressible PF xerogels were synthesised by freeze-drying using methyl-order phenolic resin as raw material, ethylene glycol as solvent, and HCl as catalyst through the sol–gel method. The superhydrophobic ASO-PF xerogels were obtained by grafting amino silanes onto the PF xerogel backbone using the Mannich reaction, with a water contact angle of 153°. The modified ASO-PF retains the pore structure (90% porosity) and excellent compressibility of the original xerogel, with a maximum compressive deformation of 80% and withstanding 10 times 50% compressive deformation without being damaged.

The grafting of silanes and the introduction of amine groups result in a high selectivity throughout the skeleton and can be applied to wastewater treatment. For waste water solutions in two-phase media (e.g., insoluble organic solvents and oil slicks) direct static adsorption can be used. The maximum adsorption capacity of chloroform, for example, is 12.98 g·g^−1^. The adsorption–desorption capacity remains stable over several cycles, while the contact angle remains at around 150°. For waste aqueous solutions in homogeneous media (e.g., phenol solutions and dyestuffs), the compression properties of the xerogel are used to propose a respiratory adsorption method to adsorption of contaminants that maximises the use of adsorption sites throughout the skeleton. For low concentration phenol solutions, the continuous diffusion of the phenol solution into the internal pores is hindered by the superhydrophobic properties of the entire ASO-PF skeleton. The aqueous phenol solution cannot enter the interior of the xerogel and is only trapped by the surface pores, resulting in a low adsorption capacity. By increasing the concentration of the phenol solution and further reducing the surface energy of the adsorbed solution, the adsorbed solution slowly penetrates into the xerogel, and then the adsorption site of the internal skeleton is maximised by squeezing and inhalation, which makes the adsorption of phenol by ASO-PF significantly more effective. For dyestuffs (Congo Red for example) this breathing method is even more effective. The unmodified PF xerogel has an affinity to the organic phase due to the entire organic skeleton. After the aqueous solution of Congo red enters the interior of the PF xerogel, Congo red is trapped in the organic skeleton and the colourless and transparent water is extruded when squeezed again, achieving a good separation effect. The modified ASO-PF, with its superhydrophobic properties, preferentially adsorbs the organic phase Congo Red from the solution during inhalation and extrusion, which also results in a good separation. As shown above, this breathing ASO-PF xerogel has a wide application in the field of wastewater treatment.

## Data Availability

The data presented in this study are available on request from the corresponding author.

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
