# Peer review of "Respiratory Adsorption of Organic Pollutants in Wastewater by Superhydrophobic Phenolic Xerogels"

_polymers, 2022, doi:10.3390/polym14081596_

Round 1
Reviewer 1 Report
Big, impressive work both in PF and ASO-PF characterization and measuring adsorption properties and modelling with Langmuir or Freundlich curve. MInor editorial mistakes to be corrected:
Line 50 - should be ..poly(vinyl alcohol)...
Line 64 - change ...Magdalena [26]... to ...Ptaszkowska-Koniarz [26].... because Magdalena this is an author first name and the last name is Ptaszkowska-Koniarz
Line 68 - change ...reach 91-118mg·g-1. ... to ...reach 91-118 mg·g-1. ....
Line 98 - change ...for 3h at... to ...for 3 h at...
Line 99 - change ....every 1h... to ...every 1 h...
Line 101 - change ...for 1.5h... to ...for 1.5 h....
Line 104 - change ...method, 4g of... to ...method, 4 g of...
Line 105 - change ...for 30min, 2g of... to ...for 30 min, 2 g of....
Line 106 - change ...for 30min and then put into the oven at 70°C for 24h to.... to ...for 30 min and then put into the oven at 70°C for 24 h to....
Line 108 - change ... every 8h, and... to ... every 8 h, and...
Line 136 - change ...test, 0.1g of.. to ...test, 0.1 g of...
Line 140 - change ... where m1 and m0 represent the mass of
the xerogel before and after adsorption. ... to ... where m0 and m1 represent the mass of the xerogel before and after adsorption. ....
Line 153 - change ...time of 1min-24h and... to ...time of 1 min-24 h and...
Line 154 - change ... 0.1g of xerogel was added to 20mL of... to ... 0.1 g of xerogel was added to 20 mL of...
Line 163 - change ... and Qe(mg·g-1) ... to ... and Qe (mg·g-1) ....
Line 225 - change ...is around 10um, ... to ...is around 10 um, ....
Line 250 - change Table 1 - add units in the top line (Temperature [0C]; Sample mass [g]; Apparent density [gcm-3]; True density [gcm-3]; Porosity [%] and skip the units from the second and third line
Line 304 - change ...Within 1s, ... to ...Within 1 s, ...
Line 306 - better describe contact angle as less than 100 than 00. On the Fig 5a clearly can be seen that CA of PF is not a 00.
Author Response
We sincerely thank you all for spending precious time and efforts in examining this manuscript and greatly appreciate your insightful comments to make the paper better. The manuscript has been carefully revised, improved and verified to address the questions raised by the reviewers, and revisions have been marked with reds for clarification. Please see the attachment.

Reviewer 2 Report
- Title: This could be improved by removing "respiratory adsorption" as this concept is not universal
- Abstract: The first sentence showed the background of the work, and the second sentence already described the material used. There should be adequate background to connect the problem and the proposed solution.
- The concept of respiratory adsorption should also be introduced briefly in the abstract
- The authors mentioned that the use of xerogels offers greater economic benefits, but there should be a clear economic analysis provided, or at least comparison with conventional adsorbents.
- The concept of inhalation-extrusion or respiratory adsorption should be introduced in the Introduction, as this was supposedly the novelty of this work.
- Fix the format of lines 167-199
- Has the aminosilane functionalization been optimized before claiming superhydrophobicity of ASO-PF?
- Line 312: Modification occurred within the pore, not "pore size within the xerogel"
- Why was the density of the organic solvents chosen as the main parameter in choosing the most suitable organic pollutant? It is supposed that functionality and chemical structure, as well as the interaction with the xerogel, would play bigger roles
- Even in the discussion section, the concept of respiratory adsorption was not mentioned or discussed, why was it then introduced in the title, like it's a very important concept?
Author Response

(The authors gave the same response as above.)
